# Sample Reweighting to Effectively Use Synthetic Data during Model Training

## Abstract

Training robust machine learning models, especially in healthcare applications, faces critical challenges due to limited labeled data, noisy labels, and class imbalance. Synthetic data generation has emerged as a promising approach to overcome these limitations. However, naively incorporating synthetic samples often introduces new challenges, such as sample quality variability and distribution mismatch. To address these issues, we propose an integrated framework that leverages Lightweight Learnable Adaptive Weighting (LiLAW) to dynamically reweigh synthetic samples based on their evolving difficulty during training. We extend LiLAW, which was developed for the multi-class classification setting, to the multi-label classification and regression settings. We then apply LiLAW to two recently introduced synthetic datasets: SynPAIN, a large-scale dataset of synthetic facial expressions designed for automated pain classification, and GAITGen, a dataset generating clinically relevant synthetic gait sequences for Parkinson's disease severity estimation. Furthermore, we validate our framework on ECG5000, a healthcare time-series dataset for heartbeat classification, with simple augmentations as well. We obtain state-of-the-art results on all of these datasets and demonstrate that LiLAW significantly improves model performance by adaptively prioritizing synthetic samples according to their difficulty. Our approach provides a computationally efficient and practical solution to improve the quality and inclusion of synthetic data in model training.

## 1 Introduction

Machine learning (ML) and computer vision (CV) systems have seen rapid advancements in healthcare, enabling more accurate diagnostics, personalized treatment, and efficient patient monitoring. However, the deployment of these systems faces major challenges due to limitations in data availability, noisy annotations, and imbalanced datasets. Collecting large-scale, high-quality, and well-balanced datasets in healthcare is particularly difficult due to privacy concerns, ethical considerations, and the logistical challenges of clinical data collection.

Synthetic data generation has been increasingly explored as an alternative to overcome these limitations. Recent advances, including SynPAIN (Taati et al., 2025), a diverse synthetic facial expression dataset for pain assessment, and GAITGen (Adeli et al., 2025), a synthetic dataset of pathological gait sequences, have helped address demographic underrepresentation and augment limited clinical data. Despite these advancements, indiscriminate incorporation of synthetic data into training often introduces issues of sample quality variability and distribution mismatch, potentially diminishing benefits or harming model performance.

To mitigate these issues, we explore the idea of adaptively weighing training samples based on their learning difficulty. Lightweight Learnable Adaptive Weighting (LiLAW) (Moturu et al., 2025), a meta-learning-based approach that adaptively prioritizes training samples by dynamically reweighing each sample's loss based on its evolving difficulty. LiLAW has demonstrated significant improvements in noisy environments and heterogeneous datasets, showing particular promise in healthcare domains where data heterogeneity and noisy annotations are prevalent. With minimal additional work, LiLAW can be integrated into existing training pipelines, requiring only three additional learnable parameters and operating without reliance on a clean validation set or extensive hyperparameter tuning. This makes it a practical and computationally efficient framework for enhancing robustness

and generalization, particularly in clinical settings where original data quality is difficult to guarantee and generated data may or may not be useful for the task at hand.

In this paper, we propose a training framework that integrates LiLAW during model training with a mixture of real and synthetic data, with the explicit goal of improving the use of synthetic samples. Rather than treating all synthetic data uniformly, we hypothesize that adaptive, difficulty-based weighting will enable more effective incorporation of synthetic samples and enhance the overall performance of the model. We validate our method using the SynPAIN and GAITGen datasets and additionally validate on ECG5000 (Goldberger et al., 2000), a healthcare time-series classification dataset with synthetic data generated from augmentations, to show that LiLAW can guide the balanced and principled use of synthetic data in practice.

Our key contributions are as follows:

- We extend LiLAW beyond multi-class classification to handle both multi-label classification and regression tasks.
- We conduct extensive validation across diverse healthcare applications, including facial pain intensity estimation (regression), gait-based Parkinson's disease severity prediction (imbalanced multi-class classification), and heartbeat abnormality classification (multi-class classification).
- We introduce an efficient strategy to make it easier to effectively incorporate and reweigh synthetic data into a given training setup.

## 2 RELATED WORK

### 2.1 SYNTHETIC DATA GENERATION

Synthetic data generation techniques address data scarcity, privacy issues, and demographic representation problems in various fields (Bauer et al., 2024). Prominent methods include Generative Adversarial Networks (GANs) (Goodfellow et al., 2014), Variational Autoencoders (VAEs) (Kingma & Welling, 2019), and diffusion-based models (Ho et al., 2020). GAN-based techniques, such as StyleGAN (Karras et al., 2018b; 2020) and Progressive GAN (Karras et al., 2018a), have successfully synthesized high-fidelity images and realistic facial data but often struggle with diversity and mode collapse issues. VAEs have also been successful in data generation with controlled latent factors but typically achieve lower fidelity compared to GANs. Diffusion models have achieved very high fidelity compared to GANs and VAEs by starting with random noise and learning to denoise to generate high-quality data. They are usually combined with text-encoders to allow for text-based prompts Zhang et al. (2024). However, they may be unable to fully capture human physiological or biological data.

Recent methods, such as SynPAIN, address these limitations by providing a diverse synthetic dataset specifically designed for pain classification, thus overcoming demographic bias prevalent in traditional datasets. Similarly, GAITGen introduces clinically relevant synthetic gait data conditioned on pathological severity, addressing the scarcity of high-quality labeled clinical data and underrepresentation of severe pathology cases. Despite these advancements, synthetic datasets need to be handled carefully due to the variability in the generation quality and distributional discrepancies with real data. These issues are not fully addressed by standard data augmentation methods.

### 2.2 ADAPTIVE WEIGHTING AND SAMPLE DIFFICULTY

Adaptive weighting methods enhance model training by dynamically prioritizing samples based on their learning difficulty or informativeness. Early methods, such as focal loss (Lin et al., 2017), addressed class imbalance by down-weighting easy samples and emphasizing harder ones. Newer approaches, like MentorNet (Jiang et al., 2018) and Learning to Reweight Examples (Ren et al., 2019), introduced more dynamic strategies but are computationally intensive or require additional clean validation sets, which may be difficult in resource-limited settings such as healthcare. To our knowledge, no methods use reweighting to make better use of synthetically generated data.

LiLAW offers a significant advancement by employing a lightweight meta-learning framework that adaptively adjusts sample weights throughout training with minimal overhead. Unlike previous meth-

ods, LiLAW does not rely on a clean validation set and uses only three learnable parameters, making it especially suitable for noisy and heterogeneous datasets commonly encountered in healthcare. This approach effectively addresses the limitations of previous methods by dynamically assessing and incorporating sample difficulty directly into training to reweigh the loss of each sample.

# 3 METHODS

## 3.1 OVERVIEW OF LiLAW

LiLAW is a method for dynamically adjusting the weight of the loss of individual training samples based on their difficulty during training. Traditional approaches to handling difficult data either discard challenging samples (Paul et al., 2021), apply fixed sample weights (Paul et al., 2021), or rely on large-scale meta-learning setups (Ren et al., 2019) that are often computationally expensive. In contrast, LiLAW introduces a lightweight, learnable mechanism that categorizes samples as easy, moderate, or hard using a simple form with three self-tuning parameters.

LiLAW operates by learning sample weights using three meta-parameters: $\alpha$, $\beta$, and $\delta$, which define the sample weighting function over the loss value. Samples with lower loss (often easy samples) are downweighed, while those with moderate loss receive higher weight, and extremely hard samples (often difficult or noisy samples) receive even higher weights. This approach follows a curriculum-learning-inspired principle, but learns the optimal weighting dynamically using gradient signals from a meta-validation step after each training mini-batch. The core contribution of LiLAW lies in its simplicity and generalizability. This makes the method suitable for plug-and-play use in any supervised learning pipeline without introducing increasing runtime or spacetime complexity much.

The method is particularly valuable in scenarios where the training dataset contains synthetic samples whose difficulty may evolve over time. For example, initially, it may be useful to focus on easy samples and later on, it may be beneficial to focus more on harder samples. In our case, synthetic datasets such as SynPAIN and GAITGen introduce varied levels of visual and structural fidelity. Without proper weighting, models may overfit to easy or unrealistic synthetic samples or underfit to clinically relevant but challenging samples. LiLAW addresses this by emphasizing moderately difficult, informative synthetic examples. Moreover, LiLAW does not require a clean validation set. Also, we note that LiLAW also improves performance not only with generated synthetic samples, but also with augmented synthetic samples, such as with the ECG5000 dataset. We evaluate multiple configurations in which the validation data may be entirely real or a hybrid of real and synthetic. LiLAW is robust across these settings and consistently learns weights that help improve downstream performance.

In summary, LiLAW is designed to be lightweight, model-agnostic, dynamic, and robust across various noise and data settings. These properties make LiLAW an ideal foundation for integrating synthetic datasets in challenging, noisy, or small-data regimes such as those found in healthcare. We use the same initial $\alpha, \beta, \delta$ values and their corresponding learning rates and weight decay values mentioned in Moturu et al. (2025).

Since LiLAW is developed for multi-class classification, it does not readily work for the cases of multi-label classification and regression. We detail how to extend LiLAW for multi-label classification and regression below.

## 3.2 EXTENSION OF LiLAW TO MULTI-LABEL CLASSIFICATION

In the $c$-class multi-label classification case, we extend LiLAW as follows:

Let $\mathcal{D}_t = \{(x_i, \widetilde{y_i})\}_{i=1}^{N}$ represent the training set and $\mathcal{D}_v = \{(x_j, \widetilde{y_j})\}_{j=N+1}^{N+M}$ represent the validation set. As with LiLAW, note that $(x_i, \widetilde{y_i})$ represents the pairs of inputs and observed (potentially synthetic/noisy) targets. For multi-label classification, $x_i \in \mathcal{X}$, where $\mathcal{X}$ is the input space and $\widetilde{y_i} \in \mathcal{Y} = \{0, 1\}^c$, is the output space with $c \in \mathbb{N}$ such that $c \geq 1$ is the total number of labels. Note that $\widetilde{y_i}[z]$ from $z = 1, ..., c$ is 0 or 1 depending on if the input has the observed label. Let $f_\theta : \mathcal{X} \to \mathcal{Y}$ be the neural network model, $\theta$ be its parameters, and $\sigma_i = \sigma(f_\theta(x_i))$ be the sigmoid of its logits.

We keep the setup similar to before, except in $\mathcal{W}_\alpha$, $\mathcal{W}_\beta$, and $\mathcal{W}_\delta$, we replace $s_i[\widetilde{y_i}]$ with $\frac{1}{k} \sum_i \sigma_i[\widetilde{y_i}]$ and replace $\max(s_i)$ with $\frac{1}{k} \sum \max_k \sigma_i$, where $k = \sum_i \widetilde{y_i}$ (total number of observed labels in $x_i$) and $\max_k \sigma_i$ obtains the $k$ max values in $\sigma_i$.

Instead of considering a single predicted value as in the multi-class classification case, we compute the average predicted probability for all $k$ observed labels corresponding to each sample. To determine the hardness of a sample, we also replace the single maximum probability with the average of the top $k$ probabilities. This is because multiple labels may be relevant to each sample. The rest of the method remains the same, but this extension ensures that LiLAW can effectively weigh samples for multi-label classification, where each sample can have multiple observed labels rather than just one.

### 3.3 EXTENSION OF LiLAW TO REGRESSION

In the regression case, we extend LiLAW as follows:

Let $\mathcal{D}_t = \{(x_i, \widetilde{y_i})\}_{i=1}^N$ represent the training set and $\mathcal{D}_v = \{(x_j, \widetilde{y_j})\}_{j=N+1}^{N+M}$ represent the validation set. As before, note that $(x_i, \widetilde{y_i})$ represents the pairs of inputs and observed (potentially synthetic/noisy) targets. However, for regression, $x_i \in \mathcal{X}$, where $\mathcal{X}$ is the input space and $\widetilde{y_i} \in \mathcal{Y} = \mathbb{R}$, is the output space. Let $f_\theta : \mathcal{X} \to \mathcal{Y}$ be the neural network model, $\theta$ be its parameters, and $f_\theta(x_i)$ be its output. Also, let $R$ be the range of the true targets, which is usually well-established in any given regression problem.

We keep the setup similar to before, except in $\mathcal{W}_\alpha$, $\mathcal{W}_\beta$, $\mathcal{W}_\delta$, we replace $\alpha \cdot s_i[\widetilde{y_i}] - \max(s_i)$ with $\frac{1}{R} \cdot (\alpha \cdot f_\theta(x_i) - \widetilde{y_i})$, replace $\beta \cdot s_i[\widetilde{y_i}] - \max(s_i)$ with $\frac{1}{R} \cdot (\beta \cdot f_\theta(x_i) - \widetilde{y_i})$, and replace $\delta \cdot s_i[\widetilde{y_i}] - \max(s_i)$ with $\frac{1}{R} \cdot (\delta \cdot f_\theta(x_i) - \widetilde{y_i})$.

In the regression case, each sample has a real-valued target, so unlike classification tasks where samples can be labeled as correct or incorrect, regression tasks measure how far the model's prediction deviates from the observed target. This deviation therefore becomes the basis for determining sample difficulty: the smaller the deviation, the easier the sample and vice-versa. Since continuous targets can span vast numerical ranges, we normalize the difference by dividing by $R$, to prevent very large or very small absolute target values from disproportionately influencing the LiLAW weighting.

### 3.4 COMPARISON OF THE 3 VERSIONS OF LiLAW

We compare the three weight functions ($\mathcal{W}_\alpha, \mathcal{W}_\beta, \mathcal{W}_\delta$) for the multi-class classification, multi-label classification, and regression cases. Note that the corresponding baseline loss function also changes accordingly.

#### 3.4.1 MULTI-CLASS CLASSIFICATION

$$\mathcal{W}_\alpha(s_i, \widetilde{y_i}) = \sigma(\alpha \cdot s_i[\widetilde{y_i}] - \max(s_i)) \tag{1}$$

$$\mathcal{W}_\beta(s_i, \widetilde{y_i}) = \sigma(-(\beta \cdot s_i[\widetilde{y_i}] - \max(s_i))) \tag{2}$$

$$\mathcal{W}_\delta(s_i, \widetilde{y_i}) = \exp\left(-\frac{(\delta \cdot s_i[\widetilde{y_i}] - \max(s_i))^2}{2}\right) \tag{3}$$

#### 3.4.2 MULTI-LABEL CLASSIFICATION

$$\mathcal{W}_\alpha(s_i, \widetilde{y_i}) = \sigma\left(\alpha \cdot \frac{1}{k} \sum_i \sigma_i[\widetilde{y_i}] - \frac{1}{k} \sum \max_k \sigma_i\right) \tag{4}$$

$$\mathcal{W}_\beta(s_i, \widetilde{y_i}) = \sigma\left(-\left(\beta \cdot \frac{1}{k} \sum_i \sigma_i[\widetilde{y_i}] - \frac{1}{k} \sum \max_k \sigma_i\right)\right) \tag{5}$$

$$\mathcal{W}_\delta(s_i, \widetilde{y_i}) = \exp\left(-\frac{(\delta \cdot \frac{1}{k} \sum_i \sigma_i[\widetilde{y_i}] - \frac{1}{k} \sum \max_k \sigma_i)^2}{2}\right) \tag{6}$$

### 3.4.3 REGRESSION

$$\mathcal{W}_\alpha(s_i, \widetilde{y_i}) = \sigma\left(\frac{1}{R} \cdot (\alpha \cdot f_\theta(x_i) - \widetilde{y_i})\right) \tag{7}$$

$$\mathcal{W}_\beta(s_i, \widetilde{y_i}) = \sigma\left(-\left(\frac{1}{R} \cdot (\beta \cdot f_\theta(x_i) - \widetilde{y_i})\right)\right) \tag{8}$$

$$\mathcal{W}_\delta(s_i, \widetilde{y_i}) = \sigma\left(-\left(\frac{1}{R} \cdot (\delta \cdot f_\theta(x_i) - \widetilde{y_i})\right)\right) \tag{9}$$

## 4 DATASETS & MODELS

### 4.1 PAIN DETECTION

**UNBC**-McMaster (Lucey et al., 2011) (we use UNBC in this paper) contains video data from 25 participants (13 females) with shoulder injuries, recorded during both painful and non-painful movements. The videos were recorded at 30 fps and had a total of 48,391 frames. Each frame is manually annotated with FACS (Ekman & Friesen, 1978) codes, allowing calculation of the Prkachin and Solomon Pain Intensity (PSPI) (Prkachin & Solomon, 2008) score. This dataset is publicly available and widely used as a benchmark for pain expression recognition research.

**UofR** (Rezaei et al., 2020) contains video recordings of 102 older adult participants, both with and without dementia. Each session was recorded at 15 fps across two conditions: a baseline lying state and an examination state where a licensed physiotherapist assisted movements to locate painful areas. After removing non-frontal frames, UofR has a total of 162,629 frames. Manual annotations were provided for 95 participants (74 females) using both PSPI and PACSLAC-II pain rating scales, of which 47 cognitively healthy older adults and 48 were residents of long-term care with severe dementia. The test results are reported on the subset of participants with dementia (**Dementia**) and without dementia (**Healthy**). Additionally, results on the entire test set are also reported (**All**).

**SynPAIN** (Taati et al., 2025) contains synthetic image pairs consisting of one neutral expression and one expressive image for each identity. Using Ideogram 2.0's commercial API, the authors generated synthetic identities along with their corresponding neutral and expressive image pairs. The dataset encompasses the following demographic and expression categories: age groups include young (20-35) and old (75+); ethnicity/race covers White, Black, South Asian, East Asian, and Middle Eastern; gender represents male and female; expression types span pain (proxy PSPI score of 1) and non-pain (proxy PSPI score of 0). Each synthetic identity comprises two corresponding images (neutral and expressive), with a total of 10,710 images across 5,355 pairs. **SynPAIN-Old** refers to the subset of SynPAIN which only contains images of the old (75+) age group. There are a total of 5,790 images across 2,895 pairs across in SynPAIN-Old.

*Model.* Pairwise with Contrastive Training (**PwCT** (Rezaei et al., 2020)) is the current state-of-the-art model for detecting painful expressions in older adults. It is trained on UNBC and UofR datasets for regression (so we use R=16 since PSPI $\in [0, 16]$, as mentioned in 3.3), but we threshold the results to obtain binary pain classification. The two key ideas of PwCT are personalized neutral baselines comparing each test expression to that person's own neutral face to reduce age-related idiosyncrasies and contrastive representation learning to improve cross-dataset generalization. The model has been externally validated in vivo and is currently being evaluated in situ. We use the pre-trained PwCT model and fine-tune it with SynPAIN in this paper using the setup described in Rezaei et al. (2020).

### 4.2 GAIT CLASSIFICATION

**PD-GaM** (Adeli et al., 2025) is a fully anonymized, publicly available 3D mesh dataset of parkinsonian gait derived from PD4T (Dadashzadeh et al., 2023). It contains 1,701 segmented walking sequences from 30 individuals with Parkinson's disease, each labeled by an expert with UPDRS-gait scores from 0 to 3 (score 4 is typically non-ambulatory). Sequences are extracted from video at 25 fps, with post-processing to correct global trajectory artifacts. PD-GaM is the largest public UPDRS-gait-annotated mesh dataset, designed to mitigate data scarcity (especially at higher severities) and to support clinically relevant evaluation metrics. We use the validation set for LiLAW's meta-validation.

**GAITGen** (Adeli et al., 2025) is a generative framework that synthesizes realistic gait sequences conditioned on Parkinson's severity (UPDRS-gait scores from 0 to 3). It disentangles motion dynamics from pathology using a Conditional Residual VQ-VAE, then generates base sequences with a Mask Transformer and refines details via a Residual Transformer, enabling precise control over impairment level. Clinician studies report near-chance discrimination between real and synthetic clips.

*Model.* We use the **MotionClassifier** (Adeli et al., 2025) model to train a lightweight supervised head on a fixed 512-D motion embeddings extracted from each gait sequence by the evaluation backbone. The classifier is a 3-layer MLP that produces class logits optimized using Adam for 7000 epochs with batch size 256 and learning rate 0.00001 when synthetic data is used and 0.0001 when synthetic data is not used. We train the MLP from scratch using the same setup as in Moturu et al. (2025).

### 4.3 HEARTBEAT CLASSIFICATION

**ECG5000** (Goldberger et al., 2000) is a five-class heartbeat dataset derived from a single 20-hour ECG tracing in the BIDMC Congestive Heart Failure Database (in PhysioNet). The preprocessing isolates individual heartbeats and interpolates each to a fixed length. 5,000 heartbeats are then randomly selected. Labels are obtained via automated annotation. We reserve 5% of the training set for LiLAW's meta-validation.

To introduce variability and simulate sensor and physiological noise, we augment 50% of the training heartbeats with additive Gaussian perturbations that combine a random offset and stochastic variance. We call this augmented dataset **ECG5000-A**. For a heartbeat sequence $x$, the augmented sample is:

$$\tilde{x} = x + b + \sigma\varepsilon, \quad b \sim \mathcal{U}[-0.2, 0.2], \quad \sigma = \sqrt{u}, \ u \sim \mathcal{U}[0.05, 0.1], \quad \varepsilon \sim \mathcal{N}(0, I).$$

Class labels are retained for augmented samples.

*Model.* We train on ECG5000 using a simple **Stacked LSTM** model (containing 2 LSTM layers) from scratch. We use the Adam optimizer for 500 epochs with batch size 512 and learning rate 0.001. We train the model from scratch using the same setup as in Moturu et al. (2025).

## 5 EXPERIMENTS & RESULTS

| Dataset configurations | | AUROC | | |
|---|---|---|---|---|
| **Training** | **Meta-validation (LiLAW)** | **Dementia** | **Healthy** | **All** |
| UofR, UNBC | - | **0.787** | 0.763 | 0.775 |
| UofR, UNBC, SynPAIN | - | 0.761 | 0.774 | 0.767 |
| | UofR | 0.767 | 0.801 | 0.784 |
| | UofR, SynPAIN | 0.778 | 0.794 | 0.786 |
| UofR, UNBC, SynPAIN-Old | - | 0.778 | 0.779 | 0.778 |
| | UofR, SynPAIN-Old | 0.782 | 0.796 | **0.789** |
| | UofR, UNBC, SynPAIN-Old | 0.763 | 0.796 | 0.780 |
| | SynPAIN-Old | 0.754 | **0.803** | 0.779 |
| | UofR | **0.784** | **0.803** | **0.793** |

Table 1: Results of fine-tuning the PwCT model with and without LiLAW under different dataset configurations (UofR, UNBC, SynPAIN, and SynPAIN-Old). The top 2 results from each column of the UofR test set (Dementia, Healthy, and All) are in **bold**.

The results on pain classification in Table 1 demonstrate the interaction between heterogeneous real datasets, synthetic data augmentations, and LiLAW within the PwCT framework. When trained only on UofR and UNBC, PwCT achieves strong baseline AUROC on the Dementia subgroup (0.787), but relatively weaker performance on Healthy participants (0.763). Incorporating synthetic data from SynPAIN or SynPAIN-Old without LiLAW gives inconsistent results: while Healthy AUROC occasionally improves, Dementia AUROC and All AUROC often stagnate or decline, suggesting

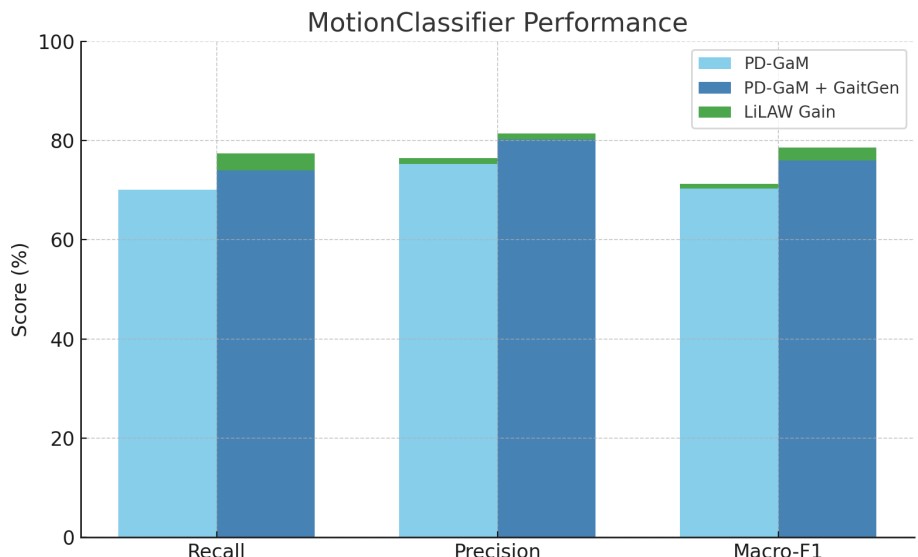

Figure 1: Recall, precision, and macro-F1 scores of the MotionClassifer with and without synthetic data and with and without LiLAW.

that naively combining real and synthetic samples risks introducing noise into model training. With LiLAW, however, meta-validation effectively reweighs real versus synthetic examples. The best and most balanced outcome is observed when training on UofR, UNBC, and SynPAIN-Old while meta-validating on UofR, resulting in AUROCs of 0.784 (Dementia), 0.803 (Healthy), and 0.793 (All). These are among the best across all settings and highlight that LiLAW selectively downweighs misleading or difficult synthetic samples. We also achieve SOTA results on the UofR test dataset. This shows that LiLAW enables PwCT to integrate synthetic data augmentation in a controlled and clinically meaningful way.

| Dataset | Recall | Precision | Macro-F1 |
|---|---|---|---|
| PD-GaM | 70.09 ↑ 0.03 | 75.32 ↑ 1.15 | 70.29 ↑ 0.98 |
| PD-GaM + GAITGen | 73.94 ↑ 3.51 | 80.20 ↑ 1.17 | 75.95 ↑ 2.60 |

Table 2: Results of the MotionClassifier model with PD-GaM and with both PD-GaM + GAITGen without LiLAW along with the increase (signified by ↑) in performance with LiLAW.

In the gait classification task, Table 2 (also see Figure 1) shows how LiLAW enhances the Motion-Classifier trained on PD-GaM and synthetic gait sequences generated by GAITGen. On PD-GaM alone, the MotionClassifier achieves recall of 70.09, precision of 75.32, and macro-F1 of 70.29. With LiLAW, each metric improves modestly, stabilizing training even without synthetic data. However, when synthetic GAITGen samples are introduced, baseline performance rises substantially to a recall of 73.94, precision of 80.20, macro-F1 of 75.95, but LiLAW amplifies these gains even further, with recall increasing by +3.51, precision by +1.17, and macro-F1 by +2.60. These results achieve SOTA for the PD-GaM test dataset. These gains are especially important clinically, as recall corresponds to sensitivity in detecting higher-severity Parkinsonian gait impairments. Rather than overfitting to synthetic distributions, LiLAW leverages them and demonstrates that synthetic augmentation paired with LiLAW can help improve subgroup generalization.

| Dataset | Acc. (%) | AUROC |
|---|---|---|
| ECG5000 | 93.60 ↑ 3.20 | 0.9982 ↑ 0.0005 |
| ECG5000-A | 96.80 ↑ 3.20 | 0.9907 ↑ 0.0093 |

Table 3: Results of the Stacked LSTM model with ECG5000 and with ECG5000-A without LiLAW along with the increase (signified by ↑) in performance with LiLAW.

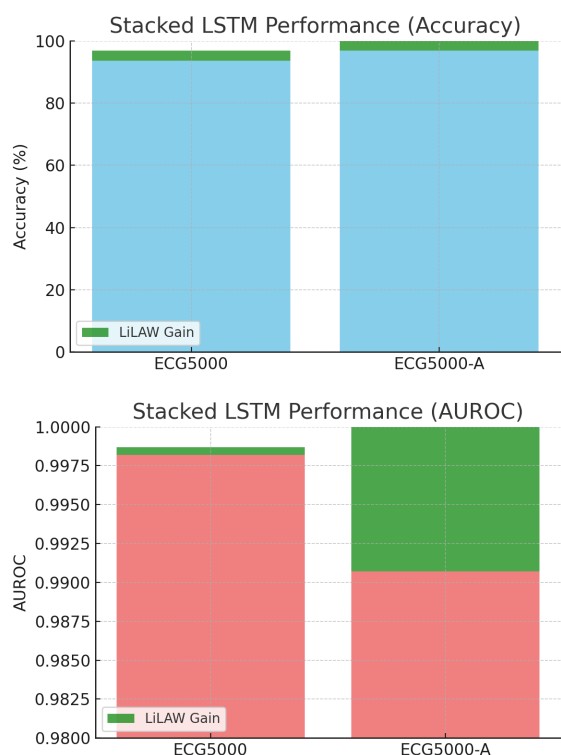

Figure 2: Accuracy and AUROC of Stacked LSTM model with and without augmented data and with and without LiLAW.

Finally, the ECG heartbeat classification results in Table 3(also see Figure 2) highlight LiLAW's utility in time-series domains where baseline performance is already saturated. Using a Stacked LSTM trained from scratch, ECG5000 alone achieves 93.60% accuracy and an AUROC of 0.9982. Even in this high-performing setting, LiLAW yields improvements (accuracy +3.20%, AUROC +0.0005). When training on ECG5000-A, which contains classical data augmentations, accuracy increases to 96.80%, but AUROC drops to 0.9907, suggesting that the augmentation increases accuracy but decreases confidence. LiLAW mitigates this imbalance by improving the AUROC by +0.0093 while preserving accuracy gains. These results achieve SOTA for the ECG5000 test dataset.

## 6 DISCUSSION

In this paper, we consider the question of whether synthetic data can be used not only as a means to increase the volume or variability of data but also as a targeted driver of generalization, when coupled with reweighting techniques such as LiLAW. Our empirical results across various real and synthetic datasets including image analysis (pain classification), movement analysis (gait classification), and time series analysis (heartbeat classification) indicate that LiLAW helps guide heterogeneous synthetic data so that performance consistently improves for downstream tasks without needing to tune hyperparameters for each task or for each domain.

**Difficulty-aware reweighting helps with synthetic data.** Naively mixing synthetic samples with real data assumes that all the data is useful and that the distributions match. In practice, synthetic examples may vary: some may be very easy to predict (risking shortcut learning), some may be informative but moderately challenging, and some may be quite hard (e.x.: mislabels). LiLAW puts weight on data points that can still help the model, while reducing the weight on those that may not be as helpful. This weight translates into higher and lower loss, respectively, for a given sample. This explains why there are performance gains with synthetic data combined with LiLAW,

but not otherwise in many cases: easy synthetic data is downweighed before oversimplifying decision boundaries, while moderately difficult, clinically plausible samples receive higher weights.

**Extendability beyond multi-class classification.** Our extensions to multi-label classification and regression generalize the domains where LiLAW could potentially used. These extensions again do not need an architectural change nor any modifications to LiLAW and its parameters.

**Simplicity.** Compared with other meta-learning frameworks, LiLAW introduces three learnable scalars and a single-batch meta-step, adding negligible runtime and spacetime. There is no dependence on a clean validation set as mentioned in Moturu et al. (2025). This low cost is crucial in healthcare, where developers may be constrained by limited compute.

**Performance improves even with dataset imbalance.** We notice that LiLAW consistently yields gains across both balanced and imbalanced regimes. By adaptively elevating the contribution of moderately hard and informative examples, LiLAW avoids the pitfalls of static weighting, i.e. get insights into fairness.

**Implications for fairness.** Because LiLAW learns adaptively, it is generally not as prone to overweighing or underweighing underrepresented groups within a dataset. This can potentially help reduce differences in subgroup performance. LiLAW may serve as a way to therefore check which subgroups may need more data.

**Limitations and future directions.** We did not explore joint training where the generator and LiLAW learn from each other. End-to-end synthetic generation with a LiLAW-informed discriminator could further improve realism and utility for the specific downstream task at hand.

## 7 CONCLUSION

Our proposed framework integrating LiLAW with synthetic datasets SynPAIN and GAITGen demonstrates substantial improvements in model performance and robustness in healthcare-oriented tasks. The dynamic adaptive weighting effectively mitigates issues associated with synthetic data integration, but also augmented data integration as shown with ECG5000. This further confirms LiLAW's broad applicability and potential to significantly enhance the practical utility of synthetic data in healthcare. This approach provides a computationally efficient, adaptive approach to optimize how to use synthetic data.

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
