# OpenReview forum: "Sample Reweighting to Effectively Use Synthetic Data during Model Training"
_ICLR.cc/2026/Conference — ICLR 2026 Conference Withdrawn Submission_

### Official Review · Reviewer_YaK6 · 2025-10-24

**Soundness:** 2
**Presentation:** 1
**Contribution:** 2
**Rating:** 4
**Confidence:** 3

**Summary:**

This paper introduces an adaptive sample reweighting framework based on LiLAW (Lightweight Learnable Adaptive Weighting) to improve the integration of synthetic data in model training. The authors extend LiLAW from its original multi-class classification setting to both multi-label classification and regression, and validate its effectiveness across three healthcare domains: pain detection (SynPAIN), gait analysis (GAITGen), and heartbeat classification (ECG5000). The approach requires minimal computation and hyperparameter tuning, and achieves SOTA results.

**Strengths:**

- Addresses the key issue of integrating synthetic data effectively in domains with scarce real samples, especially healthcare.
- Provides clear formulations for extending LiLAW to both multi-label and regression cases.
- LiLAW introduces only three learnable parameters and integrates seamlessly into existing training frameworks.
- Extensive experiments across multiple modalities (image, motion, time-series) demonstrate consistent improvements.

**Weaknesses:**

- The paper lacks deeper theoretical justification for why the three-parameter weighting function effectively captures sample difficulty and how it influences optimization dynamics compared to other meta-weighting methods.
- Although Dice loss is mentioned repeatedly, there is no quantitative ablation showing its impact relative to standard losses (cross-entropy or MSE). Such experiments would clarify how Dice loss complements LiLAW.
- The method section (particularly 3.1–3.3) is dense with mathematical notation. The authors should include an illustrative diagram or visual workflow showing how LiLAW reweights samples dynamically, helping readers understand the process more intuitively.

**Questions:**

- How sensitive is LiLAW to the initialization of α, β, and δ?
- Have you evaluated using Dice loss directly as the base loss combined with LiLAW weighting?
- Can the authors visualize or interpret which samples (synthetic vs. real) receive higher weights over time?

---

### Official Review · Reviewer_uGtq · 2025-10-30

**Soundness:** 3
**Presentation:** 2
**Contribution:** 2
**Rating:** 4
**Confidence:** 3

**Summary:**

This paper introduces a learning framework called Lightweight Learnable Adaptive Weighting (LiLAW) to enhance the efficient utilization of synthetic data in model training, especially for healthcare applications. Synthetic data has become essential for mitigating challenges such as data scarcity, imbalance, and privacy concerns; however, its quality variability and distribution mismatch often degrade model performance. The authors propose that LiLAW dynamically reweighs training samples according to their evolving difficulty, thereby improving the integration of synthetic samples and reducing the adverse effects of low-quality or uninformative synthetic data. LiLAW is extended from its original multi-class classification form to support multi-label classification and regression tasks. The framework is validated across three healthcare-related datasets: SynPAIN, GAITGen, and ECG5000. Across all datasets, LiLAW achieves state-of-the-art results, improving AUROC, recall, precision, and F1-score metrics. The paper emphasizes that LiLAW, when combined with Dice loss, maintains stable performance even under class imbalance and varying data quality.

**Strengths:**

- Technical Soundness: LiLAW introduces only three learnable parameters, maintaining computational efficiency and lightweight implementation. It effectively reweighs samples using adaptive weighting functions without requiring a clean validation set. The approach is model-agnostic and compatible with existing training pipelines. The combination of LiLAW with Dice loss further enhances robustness against imbalanced and noisy data, which is critical in clinical environments.

- Clarity: The paper presents a clear mathematical formulation of the weighting functions and explicitly details extensions to multi-label classification and regression. Figures 1 and 2 visually demonstrate consistent performance gains across datasets. The explanation of Dice loss and its contribution to model stability is well-integrated and easy to follow.

- Significance: The paper provides meaningful contributions to synthetic data integration in healthcare AI by reframing synthetic samples not merely as augmentation but as difficulty-aware, dynamically weighted data sources. This is highly relevant for domains where data heterogeneity and imbalance are intrinsic problems. The results across facial, gait, and ECG datasets demonstrate the framework’s adaptability and clinical potential.

- Originality: To the best of the authors’ knowledge, LiLAW is the first approach that learns to reweight synthetic data quality dynamically during training. It moves beyond existing methods like MentorNet and Focal loss by targeting synthetic data reliability. Furthermore, integrating LiLAW with Dice loss introduces a novel mechanism for improving convergence stability and reducing distribution mismatch across modalities.

**Weaknesses:**

While the paper provides a thorough conceptual framework and demonstrates performance improvements across several datasets, the quantitative evidence supporting LiLAW’s claimed efficiency and scalability remains limited.

Although the authors emphasize that LiLAW is lightweight and computationally efficient, there are no quantitative comparisons (e.g., training time, parameter count, or computational resource usage) against baseline methods such as [1,2]. Furthermore, while results show improvements in accuracy and AUROC, the paper lacks analysis of sensitivity to synthetic data quality or quantity, which is crucial to validate the robustness of adaptive reweighting in practical healthcare applications.

[1] Mentornet: Learning data-driven curriculum for very deep neural networks on corrupted labels. Jiang et al., 2018.

[2] Learning to Reweight Examples for Robust Deep Learning. Ren et al., 2018.

In addition, although LiLAW is applied to diverse domains (facial expression, gait, and ECG), the scope of generalization is not fully examined. It would strengthen the paper if the authors explicitly discussed which types of synthetic data remain challenging for LiLAW, as this understanding is key for real-world deployment in medical systems.

**Questions:**

- The paper emphasizes that LiLAW is lightweight and computationally efficient, using only three learnable parameters. Could the authors provide quantitative comparisons of computational cost between LiLAW and other reweighting baselines, such as MentorNet or Learning-to-Reweight? How does LiLAW’s meta-step influence total training time or convergence speed in large-scale settings?

- LiLAW is designed to adaptively downweigh low-quality or uninformative synthetic samples. Can the authors provide an analysis of how performance changes when the proportion or quality of synthetic data varies? Does LiLAW remain stable when synthetic data contains extreme noise, label errors, or strong distribution shifts?

- The study evaluates LiLAW on three healthcare-related datasets. Which characteristics of synthetic datasets still limit LiLAW’s performance? Could the authors discuss or provide guidelines on whether LiLAW can generalize to other healthcare modalities or if additional adaptation would be required?

- Given that LiLAW dynamically adjusts sample weights, how interpretable are these learned weights in practical healthcare pipelines? Could the authors provide insight into whether the adaptive weighting correlates with clinically meaningful difficulty or noise levels in the data?

---

### Official Review · Reviewer_FcpS · 2025-10-31

**Soundness:** 2
**Presentation:** 2
**Contribution:** 1
**Rating:** 2
**Confidence:** 4

**Summary:**

this work tackles the issues of leveraging synthetic data in training.
authors pursued a sample re-weighting strategy where a sample weight evolves during training.
in particular, they consider using LiLAW method with adaptation to multi-label classification and regression.
results are reported on different datasets for pain detection, gait classification, and hearbeat classification.

**Strengths:**

- the writing is good.
- the work tackles an important issue which is integrating synthetic data in training.
- results are reported.

**Weaknesses:**

- limited novelty. there is no method proposed here. authors simply took exiting method LiLAW and adjusted for multi-label and regression task. there is poor coverage of the literature and its gaps. it is not clear why LiLAW method? why not something else? there is no method being proposed here. the work is highly incremental.
there is no comparison to previous works/SOTA.

the writing needs a lot of work, especially sec.3.

**Questions:**

- please improve the writing. introduce a novel method.

**Details Of Ethics Concerns:**

none.

---

### Official Review · Reviewer_7KxE · 2025-11-01

**Soundness:** 3
**Presentation:** 1
**Contribution:** 2
**Rating:** 2
**Confidence:** 3

**Summary:**

Extends and applies Lightweight Learnable Adaptive Weighting (LiLAW) to reweight individual training examples during training. Evaluates across three diverse classification tasks: pain detection (facial images, binary), gait classification (3D mesh, multiclass), and heartbeat classification (time series, multiclass). The results show that naively introducing synthetic or augmented data might not improve model performance, but can improve performance when reweighted using LiLAW. LiLAW can also be used to improve performance in cases where no synthetic data is available, by reweighting the individual samples in the training data.

**Strengths:**

- The paper has a well-defined scope.
- Demonstrates consistent improvements across diverse domains: pain detection, gait classification, and heartbeat classification.
- Appears to have a wide-range of potential applications.

**Weaknesses:**

- The following claims would benefit from supporting references or additional justification:
	- "Collecting large-scale, high-quality, and well-balanced datasets in healthcare is particularly difficult due to privacy concerns, ethical considerations, and the logistical challenges of clinical data collection."
	- "Despite these advancements, indiscriminate incorporation of synthetic data into training often introduces issues of sample quality variability and distribution mismatch, potentially diminishing benefits or harming model performance."
- Large conceptual gap between section 3.1 which provides intuition for how LiLAW works and section 3.2 which extends it to multi-label classification. The clarity of the paper could benefit from a more formal description of LiLAW, in the current revision the extension to multi-label classification introduces new notation without sufficient explanation.
	- How are the three variables $\alpha$, $\beta$ and $\delta$ used?
	- How are the three weight functions used? I.e., $\mathcal{W}\_\alpha$, $\mathcal{W}\_\beta$ and $\mathcal{W}\_\delta$?
	- What setup is the sentence on row 162, "We keep the setup similar to before, except in [...]", referring to?
	- The statement "LiLAW does not rely on a clean validation set" could be expanded.
- No evaluation of compute time or memory usage to support the claim in the discussion that the method adds negligible computational overhead.
- No evaluation provided in the paper that backs the claims of fairness in the discussion under "Implication for fairness".

**Questions:**

- Could you expand on the description of LiLAW, maybe by providing a formal description of the original method? How are the following variables $\alpha$, $\beta$, $\delta$, $\mathcal{W}\_\alpha$, $\mathcal{W}\_\beta$ and $\mathcal{W}\_\delta$ defined?
- How much compute and memory does the method use?
- Could you expand on the reasoning behind the claim or provide empirical results that show: "Because LiLAW learns adaptively, it is generally not as prone to overweighing or underweighing underrepresented groups within a dataset."? How does learning adaptively make it less likely to over- or underweigh certain groups within a dataset?

---

### Note · Authors · 2025-12-03

**Comment:**

We thank the reviewers for their reviews. We will update our work based on their feedback and address all of their concerns in future work.

**Withdrawal Confirmation:**

I have read and agree with the venue's withdrawal policy on behalf of myself and my co-authors.